# The structural basis of promiscuity in small multidrug resistance transporters

Ali A. Kermani[1,7], Christian B. Macdonald [2,7], Olive E. Burata[3], B. Ben Koff[1], Akiko Koide[4,5], Eric Denbaum[4,6], Shohei Koide [4] & Randy B. Stockbridge [1,2 ✉]

By providing broad resistance to environmental biocides, transporters from the small multidrug resistance (SMR) family drive the spread of multidrug resistance cassettes among bacterial populations. A fundamental understanding of substrate selectivity by SMR transporters is needed to identify the types of selective pressures that contribute to this process. Using solid-supported membrane electrophysiology, we find that promiscuous transport of hydrophobic substituted cations is a general feature of SMR transporters. To understand the molecular basis for promiscuity, we solved X-ray crystal structures of a SMR transporter Gdx-Clo in complex with substrates to a maximum resolution of 2.3 Å. These structures confirm the family's extremely rare dual topology architecture and reveal a cleft between two helices that provides accommodation in the membrane for the hydrophobic substituents of transported drug-like cations.

[1] Department of Molecular, Cellular, and Developmental Biology, University of Michigan, Ann Arbor, MI 48109, USA. [2] Program in Biophysics, University of Michigan, Ann Arbor, MI 48109, USA. [3] Program in Chemical Biology, University of Michigan, Ann Arbor, MI 48109, USA. [4] Laura and Isaac Perlmutter Cancer Center, New York University Langone Medical Center, New York, NY 10016, USA. [5] Department of Medicine, New York University School of Medicine, New York, NY 10016, USA. [6] Department of Biochemistry and Molecular Pharmacology, New York University School of Medicine, New York, NY 10016, USA. [7] These authors contributed equally: Ali A. Kermani, Christian B. Macdonald. ✉email: stockbr@umich.edu

Membrane proteins from the small multidrug resistance (SMR) family are a major driver of the spread of drug resistance genes among bacteria. Genes encoding SMR proteins (variously annotated *emrE, sugE, smr, qac, ebr*) are frequently found in mobile drug resistance gene arrays, and provide a broad selective advantage by conferring resistance to ubiquitous environmental pollutants with low-grade toxicity to microbes[1,2]. The adaptive effects of the SMR proteins lead to co-selection of other genes in the arrays that confer resistance to the more potent drugs in the antimicrobial arsenal, including sulfonamides, β-lactams, and aminoglycosides, increasing the frequency of these genes in environmental reservoirs[3,4]. Thus, the dispersal of drug resistance genes among bacteria, the transport capabilities of SMR proteins, and the distribution of SMR substrates in the biosphere are intimately linked. Despite their importance, functional experiments to test the chemical scope of transported compounds have been limited to a narrow range of SMR homologs and drugs, and although the overall fold has been determined[5], sidechain-resolution structural data have not been reported for any family member. In this study, we have two objectives: (1) determine the chemical characteristics of substrates transported by representative SMR family proteins; and (2) establish the structural basis of substrate binding and transport by SMR transporters.

The sequence diversity of bacterial SMR exporters can be visualized using a sequence-similarity network (Fig. 1a and Supplementary Fig. 1). The SMR family has two major subtypes that share high sequence identity (~40%) and similarity (Supplementary Fig. 2). Both are broadly distributed across bacterial taxa, and many bacteria possess both subtypes. One group contains proteins that provide resistance against quaternary ammonium cations, including structurally diverse polyaromatic cations such as ethidium and methyl viologen. This group, the Qac cluster, includes EmrE, an *Escherichia coli* homolog

and the best-studied member of the SMR family. The other group was characterized more recently, and encompasses guanidinium (Gdm$^+$)/H$^+$ antiporters (Gdx proteins; *E. coli* gene name *sugE*)[6]. Gdm$^+$ is an endogenously produced, nitrogen-rich metabolite that is transformed or exported by genes organized in Gdm$^+$-related operons. These operons are often controlled by riboswitches that are selectively responsive to Gdm$^+$ binding[7,8] (Fig. 1a).

Initial experiments suggested that the Qac and Gdx subtypes fulfill discrete functional roles, since EmrE does not transport Gdm$^+$, and the Gdx proteins do not transport canonical EmrE drugs[6]. Of the two roles, export of quaternary ammonium ions is most readily associated with multidrug resistance, since these compounds have been used as antiseptics for almost a century[9]. But genes from the Gdx cluster also commonly colocalize with horizontal gene transfer elements (Fig. 1a)[10,11], and have been explicitly identified in mobile multidrug resistance gene arrays[12,13] (Fig. 1). Is the functional dichotomy between the Qac and Gdx subtypes as strict as early experiments suggested? Or do proteins in the SMR family share transport capabilities that make them broadly adaptive in human-impacted environments?

Here we show that SMR proteins from both the Qac and Gdx subtypes engage in promiscuous transport of hydrophobic substituted cations. Both subtypes transport a variety of hydrophobic guanidinyl compounds, and proteins belonging to the Qac subtype additionally transport substituted ammonium compounds and polyaromatic cations. X-ray crystal structures of Gdx-Clo in complex with substituted guanidinyl substrates reveal a cleft between two helices that provides accommodation in the membrane for the hydrophobic substituents of transported drug-like cations.

## Results

**Overlapping, promiscuous substrate transport by Qac and Gdx subtypes**. In order to probe the chemical characteristics of transported substrates, we performed transport experiments using exemplars of both the Qac and Gdx subtypes (Fig. 1b): the polyaromatic cation exporter EmrE, and Gdx-Clo, a functionally characterized Gdm$^+$ transporter from *Clostridales* oral taxon 876[6,14]. Radioactive uptake assays confirm that EmrE transports methyl viologen, but not Gdm$^+$; that Gdx-Clo transports Gdm$^+$, but not methyl viologen; and that both proteins discriminate against a substituted guanidinyl metabolite, agmatine (Fig. 2a).

To expand the repertoire of substrates tested, we used solid-supported membrane (SSM) electrophysiology. These experiments are feasible because the transport cycle of SMR proteins is electrogenic: the Gdx proteins couple import of two H$^+$ with export of one Gdm$^+$ ion[6], and EmrE, though it has been shown to stray slightly from strict 2:1 stoichiometry, transports monovalent substrates in an electrogenic manner as well[15,16]. In SSM electrophysiology, proteoliposomes are capacitively coupled to a gold electrode by adsorption to a lipid monolayer. When the liposomes containing SMR proteins are perfused with substrate, initiating electrogenic transport, transient capacitive currents are evoked (Fig. 2b). The peak currents are negative, consistent with a net-negative transport cycle expected for 2 H$^+$:1 substrate$^+$ exchange. (In contrast, translocation of a positively charged substrate, without concomitant proton antiport, would be expected to evoke a positive current.) The amplitudes of the currents are proportional to the initial rate of transport, but decay rapidly to baseline as a membrane potential builds up in the liposomes and the system achieves electrochemical equilibrium. Subsequent replacement of the substrate-containing solution with a substrate-free solution yields a transient current of the opposite polarity, reflecting efflux of the

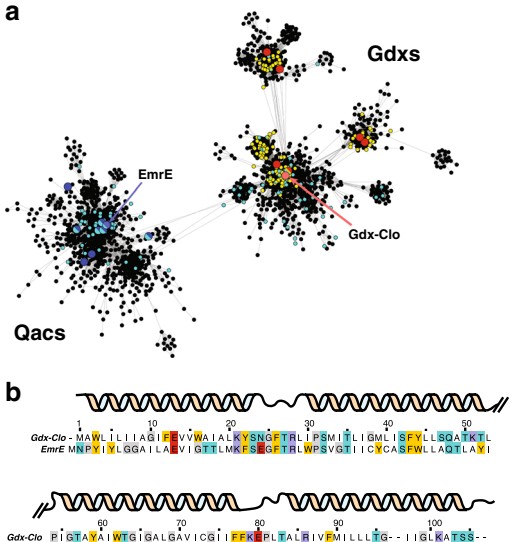

**Fig. 1 Colocalization of SMR genes with guanidine riboswitches and horizontal gene transfer elements. a** Major clusters (>10% of total set) from the sequence-similarity network of the SMR family. Each point corresponds to a cluster of sequences with >50% sequence identity. Edges between points correspond to a pairwise *E* value of at least 10$^{-20}$. Biochemically characterized proteins are shown as enlarged red (Gdx) or blue (Qac) points. SMR sequences associated with a guanidine riboswitch are colored in yellow. SMR sequences found on plasmids or genetically colocalized with integron/integrase sequences are colored cyan. Full sequence-similarity network shown in Supplementary Fig. 1. **b** Sequence alignment of EmrE and Gdx-Clo with positions of α-helices indicated.

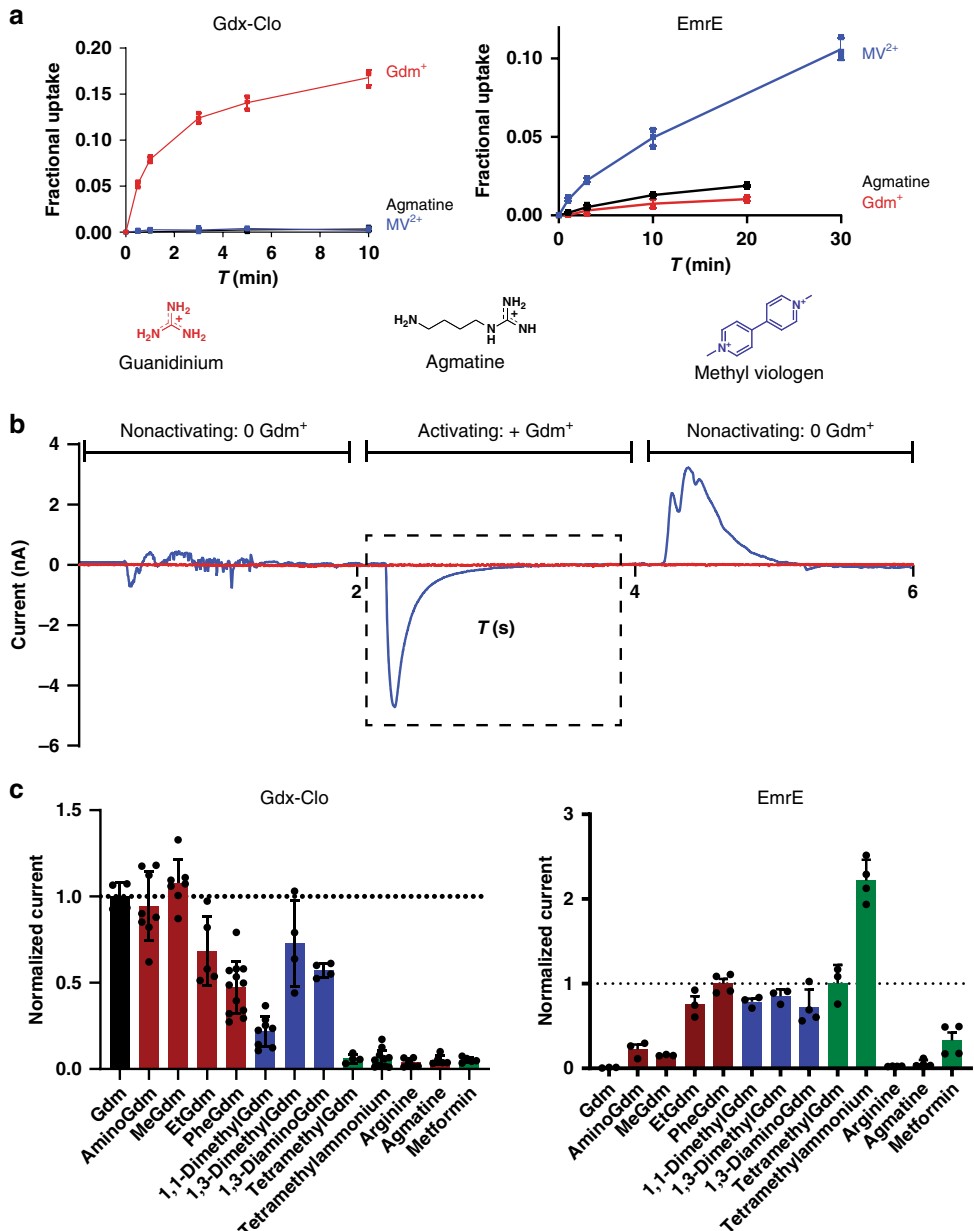

**Fig. 2 Substrate transport by Gdx-Clo and EmrE. a** Radioactive exchange assays. For Gdx-Clo and EmrE, uptake of [14]C-Gdm[+] or [14]C-methyl viologen, respectively, monitored in exchange for the indicated substrate. Points represent individual replicates; error bars represent the mean ± SEM from three independent experiments. **b** Typical SSM electrical recording of Gdx-Clo proteoliposomes perfused with indicated solutions. The area in the dashed box is used to determine initial rate kinetics. **c** Initial rate of substrate transport (peak currents) by Gdx-Clo (normalized to Gdm[+] currents, left panel) and EmrE (normalized to phenylGdm[+] currents, right panel). Singly substituted guanidinyl compounds are shown as maroon bars, doubly substituted guanidinyl compounds are shown as blue bars, with all other compounds shown as green bars. Data were collected from 3 to 4 independent sensor preparations, which were in turn prepared from 2 to 4 independent protein preparations. Individual measurements are shown as points, and error bars represent ±SEM.

accumulated substrate from the liposomes, and a return to the starting condition (Fig. 2b).

We tested substrates in the following categories: Gdm[+], guanidinylated metabolites, hydrophobic substituted guanidinium ions, and hydrophobic substituted amines. For all of these, analogous experiments with protein-free liposomes exhibit no currents (Supplementary Fig. 3). (In contrast, polyaromatic molecules like ethidium and tetraphenylphosphonium produced currents due to nonspecific partition into the membrane, and were therefore not analyzed here; Supplementary Fig. 4.) Because an unexpected shift in stoichiometry to 1 H[+]:1 substrate antiport would be electrically silent, all negative results were validated

using a second method, exchange with radiolabeled substrate (Supplementary Fig. 5 or ref. [6]). We observed no discrepancies between the electrophysiological results and the radioactive uptake experiments.

Our electrophysiology experiments (Fig. 2c) recapitulate prior observations for metabolites: EmrE does not transport Gdm[+], and both proteins are strongly selective against substituted guanidinium metabolites like arginine, agmatine, and creatine[6]. However, many of the non-natural compounds we tested were readily transported by both subtypes. Gdx-Clo transported guanidinyl compounds with hydrophobic single substitutions, including the bulky phenylGdm[+]. Currents decreased for doubly

substituted guanidinyl compounds, and were absent for tetramethylGdm$^+$. Compared with Gdx-Clo, EmrE required additional hydrophobicity and bulk in its substrates. In agreement with the radiolabeled Gdm$^+$ uptake experiments, Gdm$^+$ was not transported by EmrE. However, methyl-, ethyl-, and phenylGdm$^+$ evoked increasingly larger currents. In contrast to Gdx-Clo, EmrE also accommodated substrates with reduced or no H-bonding capacity, tetramethylGdm$^+$ and tetramethylammonium, respectively. These experiments show that polyaromaticity is not a requirement for transport by EmrE. Moreover, these experiments make clear that functional promiscuity is a general trait of the SMR family. The relative transport specificities are summarized in Supplementary Fig. 6.

**Crystal structure of Gdx-Clo.** What molecular features of SMR proteins enable these promiscuous transport functions, while simultaneously prohibiting export of endogenous substituted guanidinium metabolites? Even though this family has proved endlessly intriguing to biochemists, as one of just a few idiosyncratic examples of primitive dual topology antiparallel dimers, the only structural models available include a 7 Å electron microscopy structure of EmrE[5], and an X-ray crystal structure of EmrE[17] that has notable deficiencies: it is presented as a Cα model without sidechains, and helices that are not long enough to span the membrane and have flawed helical geometry. Computational models of EmrE that build on the low-resolution structural data have also been put forth[18,19].

In order to rectify the gap in structural information for the SMR family, we focused our crystallography efforts on Gdx-Clo. Though this protein crystallized readily, the crystals diffracted poorly. To improve diffraction, we generated monobodies, synthetic binding proteins based on the human fibronectin type III domain, to use as crystallization chaperones[20]. Upon optimization, we obtained crystals of Gdx-Clo in complex with one of the monobodies, termed Clo-L10, that diffracted to 3.5 Å Bragg spacing, and we solved the structure with phases determined by single-wavelength anomalous diffraction (SAD) of selenomethionine-substituted samples (Supplementary Table 1 and Supplementary Fig. 7). Ellipsoidal truncation of the anisotropic datasets and addition of substituted Gdm$^+$ substrates further improved resolution, ultimately to 2.3 Å. The asymmetric unit contains one Gdx-Clo dimer and two Clo-L10 monobodies, one bound to each subunit. The monobodies primarily use residues diversified in the library to bind to residues 24−32 from loop 1 of each Gdx-Clo subunit in slightly different orientations, each forming a ~400 Å$^2$ interface (Fig. 3a and Supplementary Fig. 8). In electrophysiology experiments, Gdm$^+$ currents mediated by Gdx-Clo decreased upon addition of Clo-L10, but fractional inhibition saturated at ~40%, suggesting that monobody complexation is not incompatible with the transport cycle (Supplementary Fig. 9).

**The structural basis for conformational exchange.** The two 4-TM helix subunits of Gdx-Clo are arranged antiparallel with respect to each other in non-equivalent "A" and "B" conformations. The overall fold agrees with previous low-resolution structural models of EmrE[5], and our designation of A and B subunits matches that used for EmrE. A large aqueous chamber is open to one side of the membrane, with the strictly conserved substrate- and proton-binding glutamates, E13$_A$ and E13$_B$, accessible at the bottom. Positive density is visible between the E13 sidechains, but cannot be definitively assigned as Gdm$^+$ at this resolution (Supplementary Fig. 10). Transport by the antiparallel SMR proteins involves a conformational swap between the two structurally distinct monomers, which seals the substrate

binding site on one side of the membrane while opening an identical site on the opposite side (Fig. 3b, c). As a consequence of the antiparallel homodimeric architecture, there is no structural difference between the inward-open and outward-open conformations: they are structurally identical and related by twofold symmetry about an axis parallel to the plane of the membrane. To visualize conformational exchange, we have rendered this structure in both the inward- and outward-facing directions (Fig. 3c).

The crux of the conformational exchange is helix 3 (G$_{56}$xxxAxxTG$_{64}$IGxxxAxxxG), which possesses two GxxxAxxxG helical packing motifs offset from each other by two amino acids, or just over 180°. The G$_{64}$IG sequence at the helical midpoint is the fulcrum between an N-terminal domain (TM1, TM2, and the first half of TM3) and a C-terminal domain (the second half of TM3 and TM4). Comparing subunit A and subunit B, the domains possess near structural identity (RMSD 0.5 Å for C$_α$ 1−62), but are offset by a rigid body rotation of about 30° (Fig. 3d). In agreement with our observations, the analogous G$_{64}$VG sequence in EmrE has been identified in EPR studies as a "kink" about which the conformational change occurs[21]. Inspection of the regions that change in accessibility during the transport cycle shows that, for each TM3, only one of the two GxxxAxxxG packing motifs is buried at one time, and that burial alternates with conformational exchange (Fig. 3c and Supplementary Fig. 11). We posit that competition between the two halves of TM3 to pack against structurally complementary regions of the protein contributes to the structural frustration and conformational exchange in the Gdx transporters. In addition, T63, which immediately precedes the GIG sequence, is in a position to backbond to the mainchain and further perturb the helical geometry. Mutation of the analogous serine at this position in EmrE interferes with the dynamics of the conformational exchange[22].

The well-ordered extramembrane loops also exhibit major differences in packing on the open and closed sides of the transporter (Fig. 3e). On the open side of the transporter, several charged amino acids, K21$_A$ from loop 1$_A$ and E80$_A$ and R86$_A$ from loop 3$_A$, are solvent-exposed in the aqueous chamber. Upon conformational exchange, K21$_A$, E80$_A$, and R86$_A$, converge on loop 2$_B$ and the N-terminal end of helix 3$_B$, forming cross-subunit H-bond interactions with the backbone and sidechain atoms of L53$_B$-T57$_B$. The hydrophobic loop 1$_A$ also contributes to sealing the binding pocket on the closed side of the transporter, where it is wedged between the antiparallel helices 2$_B$ and 2$_A$. Thus, the extramembrane loops, which are the least well-resolved features of previous structural models of SMR proteins, likely play an important role in the energetics of subunit packing. The involvement of loop 3 in conformational exchange has also been proposed for EmrE based on spectroscopic studies[23,24].

**The substrate binding site.** In order to visualize substrate coordination, we solved a structure of the Gdx-Clo/L10 monobody complex together with a non-natural transported substrate, phenylGdm$^+$, since this compound's bulky phenyl group would aid modeling of the substrate. Fortuitously, this also improved the resolution to 2.5 Å. We observed conspicuous density near the glutamates, to which we fit one phenylGdm$^+$ molecule (Fig. 4a). Neutralization of these glutamates has previously been shown to abolish substrate transport in Gdx-Clo[6]. The substrate's guanidinyl group is coordinated by E13$_B$, whose position is in turn stabilized by a stack of conserved H-bond donors and acceptors, including W62$_B$, S42$_B$, and W16$_B$. W62 and S42 are highly conserved among SMRs, and have been previously implicated in substrate specificity and transport[25,26]. In Gdx-Clo, mutations that remove H-bond potential, S42A and W62F, reduced or

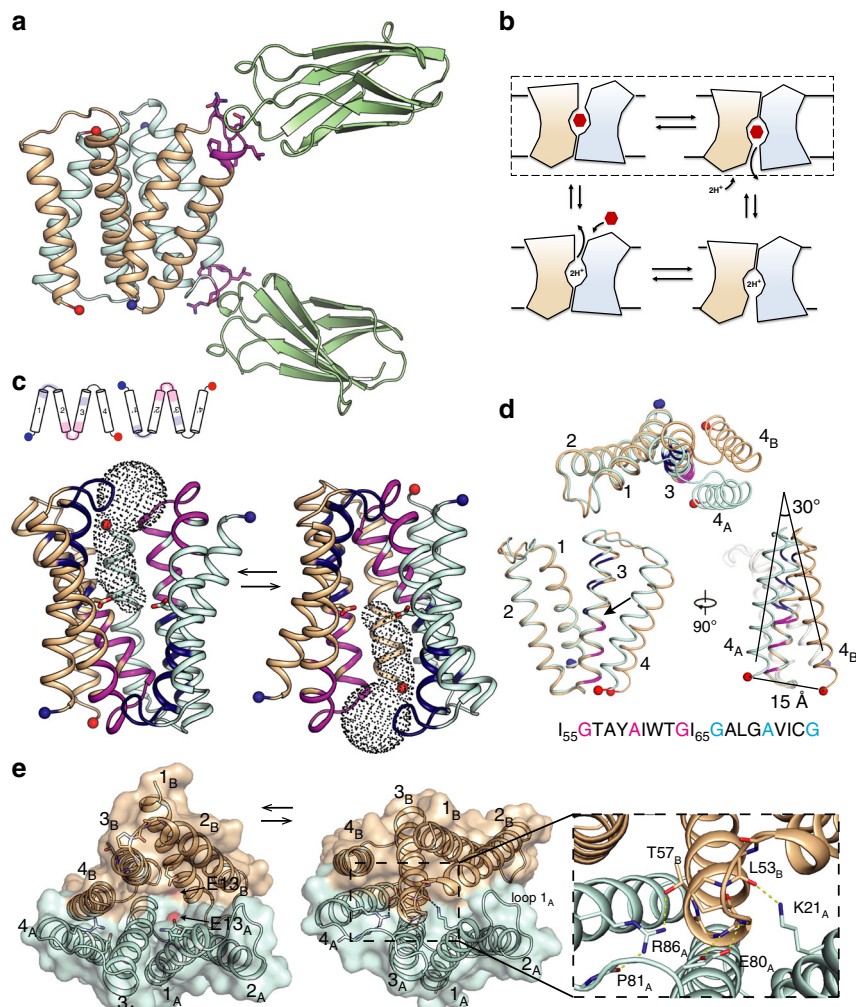

**Fig. 3 Gdx-Clo structure and conformational exchange. a** Structure of Gdx-Clo/monobody complex. Clo-L10 monobodies are shown in green. Transporter shown with subunit A in light blue and subunit B in tan. The N- and C-termini for each subunit are shown as blue and red spheres, respectively. Transporter residues that comprise the monobody binding interface are shown in magenta. **b** Cartoon schematic showing transport cycle for an antiparallel homodimer. The dashed box indicates the conformational exchange step highlighted in panel (c). **c** Changes in accessibility during conformational exchange. For both the upper diagram, and the lower three-dimensional structure, regions that alternate in solvent accessibility are shown in magenta (TM2, loop 2, and the first GxxxAxxxG motif of TM3) and dark blue (TM1, loop 1, and second GxxxAxxxG motif of TM3). The N- and C-termini are shown as blue and red spheres. In the three-dimensional structure, E13 sidechains shown as sticks and solvent-accessible vestibule indicated with dots. **d** Overlay of Gdx-Clo A and B subunits aligned over $C_\alpha$ 1−61. The sequence of TM3 is shown with GxxxAxxxG packing motifs colored in magenta and dark blue in structures and sequence. An arrow indicates $I_{65}$. Three views are shown (counterclockwise from top): top-down view, view through the plane of the membrane (with $GI_{65}G$ indicated with an arrow), and rotated 90°. **e** Conformational exchange viewed from top down. E13 sidechains shown in red as surface representation and indicated with arrows. Sidechains that make polar contacts on the closed side of the transporter are shown as sticks.

eliminated $Gdm^+$ exchange, respectively (Fig. 4b). Conspicuously, W16 is conserved among Gdx proteins, but conserved as a glycine or alanine among the Qac subtype. In Gdx-Clo, the W16G mutant reduces, but does not eliminate $Gdm^+$ exchange (Fig. 4b).

The guanidinyl group of phenyl$Gdm^+$ is also in close proximity to the opposite $E13_A$ sidechain. However, $E13_A$ is deflected downward by a cross-subunit interaction with $Y59_B$, so that the angle between the nitrogen, hydrogen (coplanar with the guanidinyl group), and oxygen atoms is not optimal for H-bond formation. Y59 is absolutely conserved among SMR proteins and the capacity to hydrogen bond has been identified as mechanistically essential at this position[18,27]. Based on our Gdx-Clo structure, we propose that $Y59_B$ and the guanidinium group compete for $E13_A$, and that displacement of $Y59_B$ by the guanidinyl group initiates the transport motion (Fig. 4c). Of all

the amino acids, Y59 undergoes one of the largest changes in conformation, swinging out away from the binding site and into the aqueous pocket when the subunits swap conformations. Y59F, which cannot form a hydrogen bond with the E13 carboxylate, is not competent for substrate exchange (Fig. 4b), in accord with the requirement for an H-bond at this position. It is also notable that E13 only forms a single hydrogen bond with the $Gdm^+$ ion. This contrasts with the lowest energy coplanar, bidentate coordination of the guanidinium/glutamate complex in solution[28], and also draws a contrast to $Gdm^+$ coordination by the guanidine riboswitches, which provide hydrogen bond partners for most or all of the substrate's five hydrogen bond donors[29–32]. The more minimal coordination by the transporter explains its permissiveness towards guanidinium ions with methyl substitutions in one or two positions.

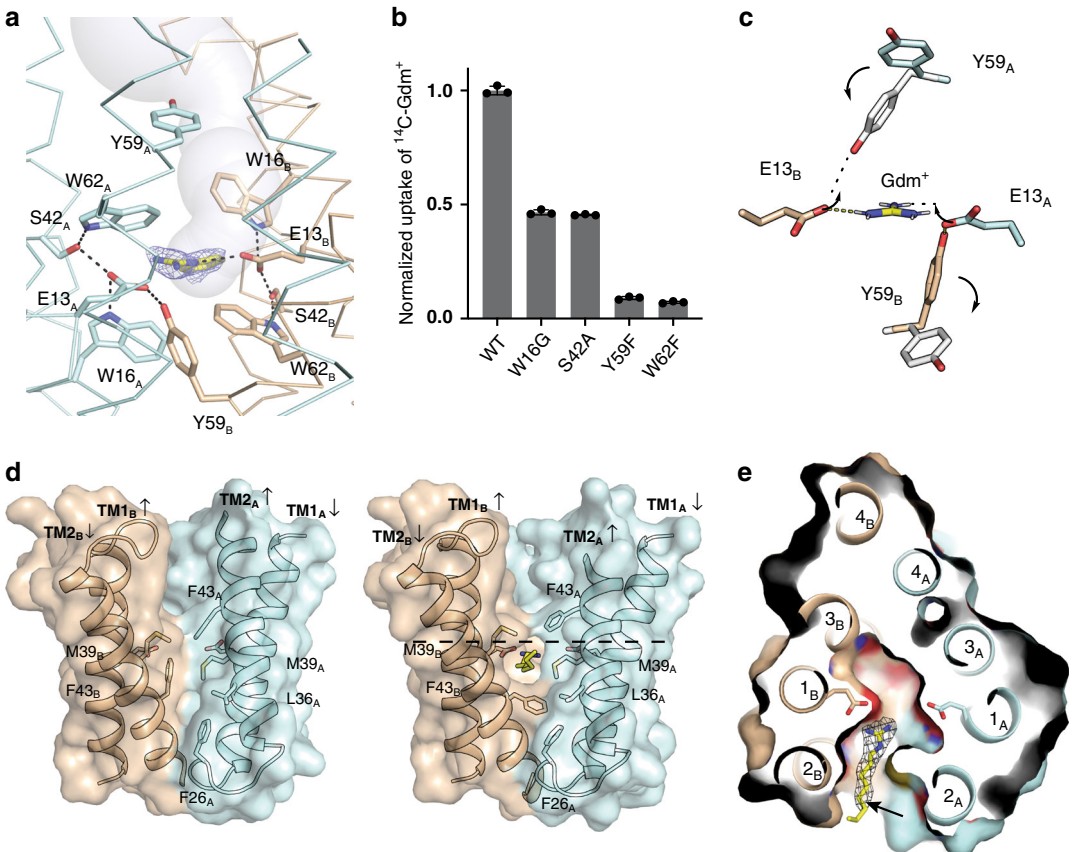

**Fig. 4 Substrate binding by Gdx-Clo. a** PhenylGdm$^+$ binding site. Subunits colored in light blue and tan as in Fig. 3. The aqueous accessible vestibule is shown as a gray surface rendering. Sidechains that coordinate substrate or E13 shown as sticks, and interactions with appropriate distance and geometry for hydrogen bonds are shown as dashed lines. The electron density assigned to phenylGdm$^+$ (2Fo-Fc map contoured at 1.3$\sigma$) is shown as blue mesh. **b** Uptake of $^{14}$C-Gdm$^+$ into proteoliposomes mediated by the indicated mutant. Total uptake is normalized relative to uptake by WT at 10 min. Error bars represent the SEM for three independent replicates. **c** Illustration of the proposed conformational transition around the transported Gdm$^+$. Colored sidechain sticks are in the positions observed in the structure; white sidechain sticks and arrows show proposed conformational change. **d** Membrane portal. The structure from Fig. 3 is shown at left, and the octylGdm$^+$ bound structure is shown at right. Cartoon is shown with helices 3 and 4 removed for clarity. Sidechains lining the portal, and E13 sidechains, are shown as sticks. OctylGdm$^+$ is shown as stick representation, with octyl tail extending toward the viewer. Dashed line indicates the level at which the protein is sliced in panel (**e**). **e** Top-down view of Gdx-Clo surface and helices with octylGdm$^+$, sliced at approximately the midpoint of the membrane. E13 sidechains are shown as sticks. Experimental 2Fo-Fc density for the ligand, contoured at 1.8$\sigma$, is shown as gray mesh. The arrow indicates C$_5$ of the octyl substituent. Agmatine and arginine bear amino groups at this position.

**A membrane portal accommodates hydrophobic substrate substituents**. In the case of phenylGdm$^+$, the substituent is packed between TM2$_A$ and TM2$_B$. At this point, the antiparallel TM2 helices splay apart, delimiting a portal from the membrane to the substrate binding site (Fig. 4d). In order to interrogate this feature, we solved a structure of Gdx-Clo in complex with octylGdm$^+$, a cationic detergent with a Gdm$^+$ head group and an eight-carbon tail. The guanidinyl group sits in the same binding pocket as phenylGdm$^+$, near E13$_B$, and the aliphatic tail protrudes from the protein and into the detergent micelle (Fig. 4d, e). The tail is accommodated by rotameric rearrangements of the hydrophobic amino acids lining TM2 including M39 and F43 (Fig. 4e). Similar portals have been observed in other drug-binding membrane proteins, and are thought to provide binding site access for hydrophobic substrates that partition into the membrane[33–35]. Spectroscopic studies and molecular modeling have provided evidence for a similar portal between the TM2 helices of EmrE[18,23].

It is clear that this membrane portal could be exploited by hydrophobic compounds to gain access to the binding site. We propose that this portal is also advantageous in the transporter's physiological context. Although this portal allows hydrophobic

substituents accommodation by the membrane, metabolites like arginine, creatine, and agmatine all have polar groups on the tails for which insertion into the hydrophobic membrane environment would introduce a high energetic penalty, making the portal a convenient means for selecting against major guanidinylated metabolites, and rationalizing the conservation of this feature. The SLC35 solute transporters[36–38] provide a notable point of contrast. SLC35 proteins are assembled as two-domain inverted repeat transporters in which each domain is homologous to the SMR fold, but have an additional two-helix insertion that seals off the portal so that the binding site is only accessible from aqueous solution (Supplementary Fig. 12).

## Discussion

In summary, our transport experiments show that a representative of the SMR family's Gdx subtype, like the better characterized Qac SMRs, promiscuously transports a series of hydrophobic non-natural compounds, and that functional promiscuity is thus a general feature of the SMR family. Although Gdx-Clo's physiological role is transport of the metabolite Gdm$^+$, it is not exquisitely selective for Gdm$^+$, and whereas there is a biological imperative to prevent export of valuable guanidinylated

metabolites like arginine or agmatine, there is no selective pressure to be discerning towards non-native compounds. Promiscuous functions, those that are not under direct selection, provide a rich source of cryptic variation that can be harnessed to provide evolutionary novelty[39]—perhaps rationalizing the broad distribution of both the Qac and Gdx subtypes with horizontal gene transfer elements. Changing environmental pressures, which could include various human-introduced biocides, may have made these latent functions adaptive. Indeed, environmental contamination by hydrophobic quaternary amines is associated with antiseptic use[9], and substituted guanidinium ions and biguanides have also been identified as widespread, long-lived, environmentally disruptive contaminants that enter the biosphere as agricultural or industrial chemicals[40–42] or pharmaceuticals that impact the human microbiome and that are excreted in waste water[43–46].

Structural analysis of Gdx-Clo reveals numerous features that correspond to biochemical or spectroscopic observations made for EmrE, indicating a high degree of mechanistic conservation between the Qac and Gdx subtypes. The Qac and Gdx subtypes also share multiaromatic binding pockets, which have also been implicated in polyspecificity in several other systems, including QacR transcriptional regulators and P-glycoprotein[33,47,48]. The structure also identifies other features that contribute to promiscuous substrate transport in the SMR family, including minimal coordination of the substrate and direct access from the membrane to the binding site. We conjecture that SMR proteins have enjoyed such evolutionary success in the modern world because this portal, a conserved selectivity mechanism against major physiological metabolites, proved to be extremely adaptive for the binding and export of hydrophobic, human-introduced chemicals.

## Methods

**Sequence-similarity network**. A sequence-similarity network was generated using the EFI-EST webserver[49] starting from PFAM family PF00893 (Multi_Drug_Res), with an alignment score of 20, and visualized with 50% similarity in Cytoscape using the prefuse force-directed layout[50]. A genome neighborhood network was then generated with the EFI-GNT tool, using a neighborhood size of 10. The coordinates of the Guanidine-I, Guanidine-II, and Guanidine-III riboswitches were retrieved from RFAM and used to annotate the SMR PFAM members if they occurred within 100 bp of an RFAM member. A set of plasmid-encoded SMRs was generated from Uniprot using the keyword plasmid. The GNN was used to annotate integron-integrase neighbors using the PFAM domains Phage_int_SAM_4 (PF13495) and Phage_integrase (PF00589).

**Transporter expression, purification, and proteoliposome reconstitution**. Lipids were from Avanti, detergents from Anatrace. Proteins were expressed and purified as previously described[6]. Briefly, Gdx-Clo bore a C-terminal hexahistidine affinity tag and a LysC recognition site, and were cloned into a pET-21c expression vector, and transformed into C41 (DE3) E. coli. When cultures reached an OD$_{600}$ of 1.0, protein expression was induced with 0.2 mM Isopropyl β-D-1-thiogalactopyranoside (IPTG) for 3 h at 37 °C. Cell lysate was extracted with 2% (w/v) decyl-β-D-maltoside (DM), and the soluble fraction was purified over cobalt affinity column, washed with 100 mM NaCl, 20 mM imidazole, and then eluted with 400 mM imidazole. The affinity tag was cleaved by incubation with LysC (200 ng per mg protein, 2 h at room temperature), before a final size exclusion purification step using a Superdex 200 gel-filtration column equilibrated in 100 mM NaCl, 10 mM 4-(2-hydroxyethyl)-1-piperazineethanesulfonic (HEPES)-NaOH, 5 mM DM, pH 8.1. EmrE was expressed and purified similarly, but the construct bore an N-terminal hexahistidine sequence with a thrombin recognition site. After induction with IPTG, protein was expressed overnight at 16 °C.

E. coli polar lipids dissolved in chloroform were dried under a nitrogen stream and residual chloroform was removed by washing and drying three times with pentane. Lipids were solubilized with reconstitution buffer (100 mM KCl, 100 mM KPO$_4$, pH 7.5) containing 35 mM 3-((3-cholamidopropyl) dimethylammonio)-1-propanesulfonate (CHAPS). For SSM electrophysiology experiments, proteoliposomes were prepared with 20 mg E. coli polar lipid per ml, and a 1:25 protein:lipid mass ratio. For radioactive flux assays and H$^+$ transport assays, proteoliposomes were prepared with 10 mg E. coli polar lipid per ml, and a 1:5000 protein:lipid mass ratio. The protein/detergent/lipid solution was dialyzed against a 1000-fold excess of reconstitution buffer, with three buffer changes over 2 days.

After the final round of dialysis, proteoliposomes were aliquoted and stored at −80 °C until use.

**Radioactive flux assays**. After reconstitution, proteoliposomes were loaded with test substrate and subjected to three freeze/thaw cycles before extrusion 21 times through a 400 nm membrane. To remove unencapsulated substrate, external solution was exchanged by passing liposomes over a Sephadex G-50 column pre-equilibrated with reaction buffer (25 mM HEPES, 400 mM sorbitol, pH 7.5). Recovered proteoliposomes were diluted twofold into reaction buffer, and the substrate transport reaction was initiated by addition of $^{14}$C-labeled compound (20 μM $^{14}$C-Gdm$^+$ for Gdx or 7 μM $^{14}$C-methyl viologen for EmrE; American Radiolabelled Chemicals, Inc., St. Louis, MO). At time points, 100 μl of reaction mixture was passed over a 1.6 ml Dowex ion exchange resin column (N-methyl-D-glucamine form), then suspended in scintillation fluid (Ultima Gold; Perkin-Elmer) for liquid scintillation counting.

**SSM electrophysiology**. SSM electrophysiology was conducted using a SURFE$^2$R N1 instrument (Nanion Technologies, Munich, Germany) according to published protocols[51]. SSM sensors were first alkylated by adding 50 μl thiol solution (0.5 mM 1-octadecanethiol in isopropanol) to a clean sensor's well, then incubating for 1 h at room temperature in a closed dish. Afterwards, the sensor was rinsed three times with ethanol and three times with water and dried by tapping on a paper towel. 1.5 μl of lipid solution (7.5 μg/μl 1,2-diphytanoyl-sn-glycero-3-phosphocholine in n-decane) was painted on the gold electrode surface using a pipette tip, followed immediately by addition of 50 μl of nonactivating buffer (100 mM KCl, 100 mM KPO$_4$, pH 7.5). Proteoliposomes were diluted 25-fold in buffer and sonicated 30–60 s before addition to the sensor surface and centrifugation at 2500 × g for 30 min.

Before experiments, sensors were checked for conductance and capacitance using SURFE$^2$R software protocols. Sensors for which capacitance and conductance measurements were outside an acceptable range (10–40 nF capacitance, 1–5 nS conductance) were not used for experiments. Sensors were periodically rechecked for quality during the course of an experiment. Each substrate was tested for transport at a concentration of 1 mM in buffer containing 100 mM KCl, 100 mM KPO$_4$, pH 7.5. For measurements in the presence of monobody, recording buffers contained 50 μg bovine serum albumin/ml. To compare measurements recorded on different sensors, currents were normalized relative to a reference compound, as described in the text. Currents elicited by the reference compound were measured both at the outset of the experiment and after collecting data on test compounds. If currents for the first and last perfusions of reference compound differed by more than 10%, this indicated that the amount of reconstituted protein had not remained stable over the course of the experiment, and data collected in this series were not used for further analysis. Data were collected from 3 to 4 independent sensor preparations, which were in turn prepared from 2 to 4 independent protein preparations. Reported data are for peak currents, which represent the initial rate of substrate transport before a membrane potential builds up and inhibits further electrogenic transport[51].

**Monobody development**. Monobody selection was performed following previously published methods[20,52,53]. Four rounds of phage selection with target concentrations of 100, 100, 50 and 20 nM was performed in 10 mM Hepes pH 7.5, 200 mM NaCl, 20 mM GdmCl, 4 mM DM, then sorted pools were subcloned into a yeast-display library following recombination of 5′ and 3′ fragments to increase library diversity[20]. Three rounds of yeast library sorting were performed: the first round for clones binding to 50 nM target, second round for clones exhibiting no binding to 10 μM streptavidin (negative sorting), and the third round for binding with 5 nM target. Isolated clones were validated for target binding using a yeast-display binding assay, as described in detail[20,54].

**Monobody expression and purification**. Monobody proteins were expressed in E. coli (BL21-DE3) grown in Studier's autoinduction media[55] 15–18 h at 37 °C. After harvesting by centrifugation, cell pellets were frozen at −80 °C for 30–45 min prior to being resuspended in breaking buffer (20 mM Tris-Cl pH 8.0, 500 mM NaCl) supplemented with 400 μg DNase, 2 mM MgCl$_2$, 1 mM PMSF, 1 mg/ml lysozyme, 25 μg pepstatin, and 500 μg leupeptin and lysed by sonication prior to centrifugation (27,000 × g for 15 min). Inclusion bodies were isolated by addition of Triton X-100 to a final concentration of 1% w/v[56], incubation of the lysate on ice, and centrifugation (27,000 × g for 15 min). The pellet containing the L10 inclusion bodies was resuspended in denaturing buffer (20 mM Tris-Cl pH 8.0, 6 M GdmCl) and incubated at room temperature with rotation for 1 h. Debris were removed by centrifugation (17,500 × g/45 min), and the supernatant was loaded onto a cobalt affinity column (Takara; 3 ml resin/l culture) for on-column refolding[57]. The column with bound monobody was washed with 10 CV of denaturing buffer, 10 CV of denaturing buffer supplemented with 10 mM imidazole, 10 CV of wash buffer (0.1% (w/v) Triton X-100, 20 mM Tris-Cl pH 8.0, 500 mM NaCl), 10 CV of refolding buffer (5 mM β-cyclodextrin, 20 mM tris-Cl pH 8.0, 500 mM NaCl), and finally, 10 CV of breaking buffer. The resin, with bound, refolded monobody, was incubated with TEV protease (0.03 mg/ml cobalt affinity resin) overnight to cleave the His$_6$ tag, and digested monobody was eluted with breaking buffer. A final size

exclusion purification step was performed using a Superdex 75 gel-filtration column equilibrated in 10 mM HEPES pH 7.5, 10 mM NaCl.

**Crystal preparation.** For X-ray crystallography, Gdx-Clo and monobody Clo-L10 were purified as described above. For the Clo purification, the size exclusion buffer contained 200 mM NaCl, 10 mM HEPES pH 8.1, and 10 mM Gdm⁺ or 20 mM phenylGdm⁺. Proteins were concentrated to 10 mg/ml, Clo-L10 was supplemented with 4 mM DM, and monobody and Gdx-Clo dimer were mixed in a 2.1:1 ratio. The protein solution was then mixed with an equal volume of crystallization solution (0.3 μl in 96-well plates). Initial hits grew in 200 mM CaCl₂, 0.1 M Tris/HCl pH 8.0 and 32.5% PEG 600. Crystals were subsequently improved by addition of charged detergents lauryldimethylamine-N-Oxide (LDAO; final concentration 6.6 mM), dimethyldodecylphosphine oxide (Apo12; final concentration 2 mM), or octylGdm⁺ (final concentration 3.3 mM) to the protein solution prior to admixture with the crystallization solution (0.45 μl protein/detergent mixture together with 0.3 μl crystallization solution). Optimized crystals typically grew to their maximum size in 14 days in a wide range of salt and pH conditions with 32–36% PEG 600. For selenomethionine-incorporated protein, the best diffracting crystals were obtained with Apo12 supplementation, and crystallization solution 0.1 M LiNO₃, 0.1 M N-(2-Acetamido)iminodiacetic acid (ADA) pH 6.8, and 35% PEG 600. For phenylGdm⁺ bound protein, the best diffracting crystals were obtained with Apo12 supplementation, and crystallization solution 0.1 M LiNaSO₄, 100 mM Tris pH 8.8 and 34% PEG 600. For the octylGdm⁺ bound structure, octylGdm⁺ was used as the detergent additive, and crystallization solution contained 0.1 M calcium acetate, 0.1 M HEPES pH 7.5 and 33% PEG 600. Crystals were frozen in liquid nitrogen before data collection at the Life Sciences Collaborative Access Team beamline 21-ID-D at the Advanced Photon Source, Argonne National Laboratory.

**Structure determination.** Diffraction data were collected at an X-ray wavelength of 0.978 Å for selenomethionine-labeled crystals. Diffraction data were processed and scaled using DIALS[58]. The space group for the initial crystals was determined to be C121 with one Clo dimer and two monobodies per asymmetric unit. Eight selenium sites were located using SAD implemented in SHELX[59]. The positions were refined and initial phases were calculated using SHARP[60] with solvent flattening with SOLOMON[61]. A model for the transporter was built into experimental electron density maps using Coot[62]. The L10 monobodies were modeled based on a previously determined structure of a loop-library monobody (PDB code: 5NKQ [https://www.rcsb.org/structure/5NKQ])[63]. Variable loops were not included in the monobody model. These models were placed into the experimental electron density maps using Phaser-MR[64]. Partial models were cycled back into SHARP for phase calculation to improve the initial solvent envelope. Density from both the sidechains and the monobody loops was clearly visible in the electron density maps, and loops and the transporter's amino acid sidechains were built using the Se sites to ensure the correct register, with iterative rounds of refinement in Refmac[65] with prior phase information incorporated as Hendrickson−Lattman coefficients. Model validation was carried out using the Molprobity server[66].

Diffraction resolution was improved in subsequent datasets upon the addition of phenyl- or octylGdm⁺. With phenylguanidinium as the substrate, proteins crystallized in C121 as before, and with octylGdm⁺ as the substrate, proteins crystallized in P1. The arrangement of proteins in the crystal lattice was highly similar to the C121 crystal form, but with two transporters and four monobodies per asymmetric unit. Crystals diffracted anisotropically, and electron density maps were improved by anisotropic truncation of the unmerged data using the Staraniso webserver[67] with a cutoff level of 1.8 for the local $I/\sigma \langle I \rangle$. Models were built into experimental density maps calculated from Phaser, with the initial models of Gdx-Clo and L10 monobody determined previously, with iterative rounds of refinement in Phenix and Refmac. The structural model was revised in real space in Coot. Solvent-accessible vestibules were visualized with CAVER[68].

**Reporting summary.** Further information on research design is available in the Nature Research Reporting Summary linked to this article.

## Data availability

Atomic coordinates for the crystal structures have been deposited in the Protein Data Bank under accession numbers 6WK5 for Gdx-Clo, 6WK8 for Gdx-Clo bound to phenylguanidinium, and 6WK9 for Gdx-Clo bound to octylguanidinium. Materials are available upon reasonable request. Source data are provided with this paper.

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

## Acknowledgements

We are grateful to Andre Bazzone for technical advice about SSM electrophysiology and to the beamline staff at LS-CAT for assistance with crystallography data collection. We thank Ming Li, Chris Miller, and Shimon Schuldiner for helpful comments on the manuscript. This work was supported by a SURFE$^2$R N1 research grant (Nanion Technologies) and National Institutes of Health grants R35 GM128768 to R.B.S. and R01 CA194864 to S.K. This research used resources of the Advanced Photon Source, a U.S. Department of Energy (DOE) Office of Science User Facility operated for the DOE Office of Science by Argonne National Laboratory under Contract No. DE-AC02-06CH11357. Use of the LS-CAT Sector 21 was supported by the Michigan Economic Development Corporation and the Michigan Technology Tri-Corridor (Grant 085P1000817). R.B.S. is a Burroughs Wellcome Fund Investigator in the Pathogenesis of Infectious Disease.

## Author contributions

Conceptualization, A.A.K., C.B.M., and R.B.S.; Methodology, A.A.K., C.B.M., A.K., S.K., and R.B.S.; Validation, O.E.B.; Formal analysis, A.A.K., C.B.M., and R.B.S.; Investigation, A.A.K., C.B.M., O.E.B., B.B.K., A.K., and E.D.; Writing—original draft, A.A.K., C.B.M., and R.B.S.; Writing—review and editing, A.A.K., C.B.M., O.E.B., A.K., S.K., and R.B.S.; Visualization, A.A.K., C.B.M., O.E.B. and R.B.S.; Supervision, R.B.S.; Project administration, R.B.S.; Funding acquisition, S.K. and R.B.S.

## Competing interests

A.K. and S.K. are listed as inventors for patents (US9512199 B2 and related patents and applications) covering aspects of the monobody technology filed by the University of Chicago and Novartis. A.A.K., C.B.M., O.E.B., B.B.K., E.D., and R.B.S. declare no competing interests.
