## [Peer Review File · Nature Communications]

REVIEWER COMMENTS

Reviewer #1 (Remarks to the Author):

The SMR transporters display a broad specificity and provide resistance against various toxic compounds by active removal in an H⁺-coupled mechanism. More recently, Stockbridge's lab has characterized a cluster of transporters within this family that specifically remove guanidinium (Gdm⁺) cations and display much lower promiscuity. They report the X-ray structure of Gdx-Clo, a functionally characterized Gdm⁺ transporter from Clostridiales oral taxon 876. This is a long-awaited structure at high resolution and provides essential information to understand the mechanism and promiscuity of this important family of transporters.

In the first part of their work, Stockbridge and collaborators carefully characterize the substrate specificities of each cluster's representative: the GDX-Clo from the new cluster and EmrE, an *E. coli* homolog from the Qac cluster and the best-studied member of the SMR family. Most of the substrates are tested using solid-supported membrane electrophysiology, a relatively new and powerful tool to study electrogenic transport. The results confirm previous observations: EmrE does not transport Gdm, and both proteins are strongly selective against substituted guanidinium metabolites like arginine, agmatine, and creatine. The technique has limitations, and it cannot be used with hydrophobic polyaromatic substrates such as ethidium. I am surprised that Methyl Viologen produced currents in protein-free-liposomes because it is supposed to be quite hydrophilic, and maybe the currents are reflecting nonspecific interactions with lipids or with the detector. Another limitation of the technique is with very slow substrates such as TetraPhenylphosphonium, the classical substrate of EmrE. It cannot detect electroneutral events, and the authors confirmed basic observations using exchange with radiolabeled substrates.

The beautiful structure of Gdx-Clo with substrate identified interactions with highly conserved residues among the SMRs that have been previously implicated in substrate specificity and transport. In this context, regarding the highly conserved Y59 (Y60 in EmrE), they should cite the original article by Rotem et al. (*J. Biol. Chem* 281, 18715 (2006)). In EmrE, Ser and Thr partially replace Tyr in the transport assay, and Cys and Phe partially support binding. I do not know if this is the case in GDX-Clo, but maybe they can comment on this finding.

I am curious whether the authors made a homology model of EmrE and tried to explain the difference in specificities? All the residues that interact with Phenyl Gdm are conserved in EmrE, but still, EmrE does not recognize it as a substrate or, at least, an inhibitor? After all, tetramethyl guanidinium and the non-bulky metformin seem to be a substrate of EmrE (Fig. SX). The authors may already be working on this issue, and I am looking forward to the results of their studies. In any case, it would be useful to comment on this issue already here.

Since the emphasis of this paper is on substrate promiscuity, it would be interesting to comment on the similarity of the binding site to that observed in other multidrug transporters with overlapping specificities. I think that such an aromatic cage in the binding site as documented here is not limited to SMRs and similar solutions are seen in other multidrug transporters and even in soluble transcription factors such as QacR (Murray et al. (*J Biol Chem* 279, 14365-14371 (2004) and Peters et al. (*PLoS One* 6, e15974 (2011))).

Reviewer #2 (Remarks to the Author):

This is a mostly well-executed functional and structural investigation of a SMR transporter that provides a clearer structural framework of dual-topology proteins; an area that has been highly debated over the years.

Major concerns.

1. Glu14 in EmrE is not only critical for proton coupling, but its protonation states further dictates the rate of conformational exchange between inward- and outward-facing EmrE. Although it has been shown that Gdx-Clo is an electrogenic transporter, as far as I am aware, it has not been shown that the mutation of the conserved glutamic acid (E13) in Gdx-Clo abolishes proton-coupled transport. Given the interaction shown here between E13 and Gdm⁺ in the crystal structure, it makes perfect sense as the proton carrier. However, proton coupling is not discussed in the paper and nor is the mutation of E13A to alanine carried out. Since proton coupling is a key mechanistic question, I find it very odd that it is not even mentioned in the paper ... which of course makes me additionally curious why it is not?

Minor points.

1. To avoid proton leakage the transporter has to adopt an occluded, intermediate conformation, a critical step in its rocker-switch alternating-access mechanism. In the semi-sweet protein's - that form homoparallel dimers from 3 TMs - a very clear occluded conformation is formed by rearrangement of both halves of the protein. Please clarify the postulated occluded state for Gdx-Clo.
2. The semi-sweets are very dynamic and even in unbiased MD simulations it is possible to capture all major conformations. Presumably Gdc-Clo is also very dynamic and is stabilised by the monobodies. Is this that correct?

Reviewer #3 (Remarks to the Author):

The manuscript by Kermani et al, "The structural basis of promiscuity in Small Multidrug Resistance Transporters" describes two main findings, First evidence that the Gdx subfamily of SMR transporters still retain some substrate promiscuity and are not highly selective for only guanidinium, and Second the structure of a Clostridiales Gdx transporter alone and in complex with phenyl- or octyl-guanidinium. This work is important because it sheds light on a number of structural details of SMR-family transporters that have been speculated on the basis of prior biophysical studies and low-resolution structures. The insights into conformational exchange, an important mechanistic step, and how poly-specific substrate binding is achieved are novel and of broader interest for multidrug recognition and transporter function more broadly, making this work significant and appropriate for the broad Nature Communication audience.

There are several points where I would appreciate clarification in the manuscript:

1. The authors discuss subunit A and B. Is this the same A/B designation as previously labeled for the EmrE crystal structure and solution and solid-state NMR studies? It is not entirely clear to me based on the discussion of asymmetric substrate binding whether this matches the asymmetric binding reported based on NMR data for EmrE?
2. Solid-supported membrane electrophysiology appears to be a powerful tool for assessing the substrate specificity of this family of transporters. The authors provide significant detail describing the experimental setup, but much less information on how the data is analyzed. Is the initial rate the slope of the initial current under activating conditions? From my understanding of this method, it appears that peak current is commonly used because this apparently reflects initial transport before opposing current builds up to oppose transport. Are the authors comparing peak current (described as initial rate for the reason above), if so this could use clarification. If not using peak current, could some justification be provided for why a different analysis method was chosen? I expect SSME is not familiar to all readers of Nature Communication and clarification of this powerful technique would be

valuable.

3. On a related note, could the authors explain the very different timescales (minutes versus seconds) for the two sets of assay data in Fig. 2a and 2b? Why is radioactive uptake observed over minutes, while SSME apparently detects transport on a second timescale? How does this compare to the timescale other SMR transport assays in the literature?

4. Although the functional data indicates that Gdx is not highly selective, all of the substrate appear to be guanidinium derivatives. Thus, it seems to be much less promiscuous than the Qac subfamily typified by EmrE. Is it possible from this data to define the relative promiscuity of these two transporters?

Assuming that these points can be addressed, I would recommend this manuscript for acceptance due to its significance in the fields of transporter biology and multidrug recognition.

Reviewer #4 (Remarks to the Author):

Understanding the underlying mechanisms for substrate selectivity in transporters contributing to multidrug resistance is very important. Proteins of the SMR family are one of the most important factors in drug resistance among bacteria. The authors compared two transporters – EmrE and Gdx - representing the two different subtypes within the SMR family. From literature, both subtypes seem to show distinct functional characteristics and no overlap in substrate specificity.

The authors addressed two objectives: Characterization of substrate specificity and determination of the substrate binding sites structure. Both together allow for important conclusions about the structure-function relationship of SMR proteins.

First, the authors confirmed the distinct substrate specificities of EmrE and Gdx using radioactively labeled compounds. They then switched to SSM-based electrophysiology for further functional characterization, since SMR proteins are electrogenic and this technique evades the need radioactively labeled substrates.

The interpretation of their results is straight forward and conclusive: The currents resulting from activation by substrate concentration jumps show negative amplitudes, reflecting the net charge translocation of one positive charge out of the proteoliposomes (2H⁺ against 1 substrate⁺). The authors explained that the current amplitudes reflect transport of the respective substrates. Since in SSM-based electrophysiology all recorded currents (transport as well as pre steady-state currents) are transient due to capacitive coupling, this conclusion is very often tricky. But in the case of the assay performed by the authors, this conclusion is perfectly valid: Activation by positive charged substrates yielded negative currents. This can only be explained by completion of a whole transport cycle which includes the efflux of two protons and yields a net translocation of one positive charge out of the proteoliposomes. Binding or pre steady-state currents would most likely generate a positive charge translocation due to the positive charge within the substrates. In addition, the transient currents show relative broad shapes; in most cases with half widths > 200 ms (estimated from the figures), a typical indicator for transport currents.

The authors also performed important control experiments: 1) Since some compounds interact with the lipid sensor and may generate false positive currents, the authors performed the same experiments using empty liposomes. They found that polyaromatic molecules interact with the lipid sensors and excluded them from their analysis. 2) To exclude false negative results due to possible electroneutral transport modes, they analyzed all compounds showing no currents on the SSM using radioactive uptake assays. There were no discrepancies between electrophysiology and radioactive uptake experiments.

This study confirmed some known differences in substrate specificities of EmrE and Gdx, but also highlighted an overlap. In addition, it proved that polyaromaticity is not a requirement for transport by

EmrE, which is an important finding. Instead, functional promiscuously was described as a general feature of the SMR family.

To make this an even more complete story, the authors aimed for structural information on the binding pocket of Gdx. Using a crystallization chaperone, they achieved a resolution of 2.3 Å. And they also showed that monobody binding does not block the transport function by performing the electrophysiology assay in presence of monobody. The structure obtained therefore most likely represents a functional transporter state. In addition, the overall fold agrees with previous low-resolution models of EmrE. To conclude information about substrate-transporter interactions, the authors solved the structure in presence of a substrate. In addition, they found a membrane portal in Gdx likely providing binding site access for hydrophobic substrates that partition into the membrane.

The authors conclusions from the structural and functional data about the mechanism of substrate binding and translocation is clearly written and easy to follow. In addition, the authors supplemented their interpretation by various cross references within literature.

In summary, the manuscript makes a complete story filling the gaps of missing functional as well as structural information about SMR proteins. The functional data was validated by two different methods, which further enhances their value. The method section contains a complete description of the applied techniques and allows reproducing the data in other labs. In addition, SSM-based electrophysiology – a rather new technique – was applied properly: Control experiments have been performed and the experiments have been repeated several times using different protein and sensor batches. I would highly recommend the manuscript for publication in Nature Communications.

Andre Bazzone

We thank the reviewers for carefully reading our manuscript and providing helpful comments on the clarity and presentation. We have addressed each comments as indicated below:

In response to reviewer 1:

1) In this context, regarding the highly conserved Y59 (Y60 in EmrE), they should cite the original article by Rotem et al. (J. Biol. Chem 281, 18715 (2006). In EmrE, Ser and Thr partially replace Tyr in the transport assay, and Cys and Phe partially support binding. I do not know if this is the case in GDX-Clo, but maybe they can comment on this finding.

We thank the reviewer for noticing the omission of this reference and have incorporated a comment about the importance of hydrogen-bonding capacity at this position in EmrE.

“Y59 is absolutely conserved among SMR proteins and the capacity to hydrogen bond has been identified as mechanistically essential at this position^{18,27},”

2) I am curious whether the authors made a homology model of EmrE and tried to explain the difference in specificities? All the residues that interact with Phenyl Gdm are conserved in EmrE, but still, EmrE does not recognize it as a substrate or, at least, an inhibitor?

We think we've confused the reviewer on this point. Phenyl Gdm⁺ is in fact transported, as shown in Figure 2. We've endeavored to make this sentence clearer, replacing

“Whereas Gdm⁺ was not transported, methyl-, ethyl-, and phenylGdm⁺ evoked increasingly larger currents”

with

“In agreement with the radiolabeled guanidinium uptake experiments, Gdm⁺ was not transported by EmrE. However, methyl-, ethyl-, and phenylGdm⁺ evoked increasingly larger currents.”

With regards to the homology model, our model corresponds well with the MD model by Vermaas...Rempe & Tajkhorshid (PNAS 2018), including the correspondence of functionally essential and conserved sidechains. Until a better experimental structural model of EmrE is determined, we think that the Vermaas et al MD model is a better resource than any homology model we might come up with!

3) It would be interesting to comment on the similarity of the binding site to that observed in other multidrug transporters with overlapping specificities.

We agree that this is interesting to consider. Since the Qacs are more deeply involved in polyaromatic transport, we think this is especially worth analyzing when a structure of EmrE is finally obtained. But it's definitely worth pointing out, and we have added the following comment to the discussion:

The Qac and Gdx subtypes also share multiaromatic binding pockets, which have also been implicated in polyspecificity in several other systems, including QacR transcriptional regulators and P-glycoprotein.^{33,44,45}

4) I am surprised that Methyl Viologen produced currents in protein-free-liposomes because it is supposed to be quite hydrophilic

The reviewer is quite right. We've corrected to text to read that ethidium and **TPP+** showed currents in protein-free-liposomes, and also added Supplementary Figure 4 to show our data for this. We didn't analyze methyl viologen because it is divalent and transported electroneutrally by EmrE (Rotem, Schuldiner, 2004, JBC).

In response to reviewer 2:

Major concern:

Proton coupling is not discussed in the paper and nor is the mutation of E13A to alanine carried out. Since proton coupling is a key mechanistic question, I find it very odd that it is not even mentioned in the paper which of course makes me additionally curious why it is not?

Thanks for pointing out this oversight. We'd mutated the glutamate in the Gdx protein in a previous manuscript (Kermani et al, PNAS, 2018), and the results were unsurprising: in radioactive uptake assays, E13Q abolishes Gdm+ uptake. Although we have not specifically examined the E13A mutation, the result with E13Q provides functional evidence that E13 behaves analogously to its counterpart in EmrE. We've included a comment and citation in the text:

“Neutralization of these glutamates has previously been shown to abolish substrate transport in Gdx-Clo⁶.”

Minor points:

Please clarify the postulated occluded state for Gdx-Clo.

We're working with some MD experts to try and understand the conformational exchange and the nature of the occluded state, but we have no information on that right now, and we believe it is beyond the scope of this work.

Presumably Gdx-Clo is also very dynamic and is stabilised by the monobodies. Is this correct?

While it is certainly possible that the monobodies stabilize a particular conformation, we show in supplementary figure 7 that monobody binding does not completely inhibit substrate transport, indicating that essential dynamics are not abolished. More likely, the monobody promotes crystallization by increasing the polar surface area available for crystal packing.

In response to reviewer 3:

1) Is this the same A/B designation as previously labeled for the EmrE crystal structure and solution and solid-state NMR studies?

Yes, we have named the subunits to agree with the EmrE literature. We have added a comment in the text to point this out.

2) Solid-supported membrane electrophysiology: From my understanding of this method, it appears that peak current is commonly used because this apparently reflects initial transport before opposing current builds up to oppose transport. Are the authors comparing peak current (described as initial rate for the reason above), if so this could use clarification.

Yes, we are comparing the peak currents. We have made this more explicit in the main text, Figure 2 legend, and the Materials and Methods to clarify that the bar graph represents peak currents.

3) Why is radioactive uptake observed over minutes, while SSME apparently detects transport on a second timescale? How does this compare to the timescale other SMR transport assays in the literature?

As we describe in the methods, the SSM experiments rely on saturating substrate (1 mM) and high protein concentrations. The radioactive exchange assays use 20 μM substrate, which is well below the $\sim 600 \mu\text{M}$ K_m (radiolabeled Gdm⁺ is expensive.), along with low amounts of protein (~ 1 protein/liposome). Both the protein and substrate concentration contribute to the slower bulk response in the radioactive uptake assays. We hesitate to make comparisons to the EmrE literature, since SSM has not been previously applied to SMR proteins, SSM provides only a rough estimate of the unitary transport rate, and in any case, the panel of substrates we tested is completely different (most NMR dynamics experiments utilize TPP⁺ and derivatives). The other major assay in this area is radioactive exchange – we can confirm that methyl viologen exchange is of the same order as in previous experiments in the literature (Yerushalmi...Schuldiner 1995, JBC).

4) Although the functional data indicates that Gdx is not highly selective, all of the substrate appear to guanidinium derivatives. Thus, it seems to be much less promiscuous than the Qac subfamily typified by EmrE. Is it possible from this data to define the relative promiscuity of these two transporters?

Yes, our data show that Gdx-Clo is promiscuous among substituted guanidinyll derivatives, but does not transport other families of compounds including quaternary ammoniums and primary amines (shown in Figure 2). We have added Supplementary Figure 6 to visualize the relative transport specificities of the two subtypes.

In response to reviewer 4:

We thank the reviewer for their positive review!

REVIEWERS' COMMENTS

Reviewer #1 (Remarks to the Author):

The authors have satisfactory responded to my comments.

Reviewer #2 (Remarks to the Author):

Thank you for clarifying that the E13 mutant abolished proton-coupled transport as expected; indeed, I had looked at the PNAS paper, but missed the Fig2A uptake as it was labelled Clo-E13A and had a number of transport curves on the same figure. Will be interesting to see how transient the occluded-state is and whether it can be populated spontaneously in unbiased MD simulations when E13 is protonated and substrate is bound.

Great paper!